# Physical Activity in Cerebral Palsy: A Current State Study

**DOI:** 10.3390/healthcare12050535

**Published:** 2024-02-23

**Authors:** Alberto J. Molina-Cantero, Thais Pousada García, Soraya Pacheco-da-Costa, Clara Lebrato-Vázquez, Alejandro Mendoza-Sagrera, Paolo Meriggi, Isabel M. Gómez-González

**Affiliations:** 1Departamento de Tecnología Electrónica, ETS Ingeniería Informática, Universidad de Sevilla, Avda de Reina Mercedes sn., 41012 Sevilla, Spain; almolina@us.es (A.J.M.-C.); clebrato@us.es (C.L.-V.); 2TALIONIS Research Group—CITIC, Universidade da Coruña, 15071 A Coruña, Spain; thais.pousada.garcia@udc.es; 3Neuromusculoskeletal Physical Therapy in Stages of Life Research Group (FINEMEV), Physical Therapy Degree, Department of Nursing and Physical Therapy, Faculty of Medicine and Health Sciences, Universidad de Alcalá de Henares, Autovía A2, km 33.200, 28805 Alcalá de Henares, Spain; soraya.pacheco@uah.es; 4Physical Therapy Department, ASPACE Sevilla, 41704 Dos Hermanas, Spain; aspacesevilla@aspacesevilla.org; 5IRCCS Fondazione Don Carlo Gnocchi, Via Capecelatro 66, 20148 Milano, Italy; pmeriggi@dongnocchi.it

**Keywords:** physical activity, cerebral palsy, disability, rehabilitation, exercise

## Abstract

This document analyzes a survey conducted in three geographical areas in Spain, focusing on centers for individuals with cerebral palsy (CP). The study aims to determine the adherence rate to recommended physical activity guidelines, assess if there is a decline in interest in physical activity over time, identify the stage at which this decline occurs, and explore potential mechanisms, tools, or strategies to sustain long-term engagement in regular physical activity for this population. The 36-item questionnaire comprises multiple-choice, open-ended, and Likert scale-type questions. Data were collected on physical activity frequency and duration, daily living activities, and demographics. Statistical analysis identified patterns and relationships between variables. Findings reveal that only a 17.6% meets the World Health Organization (WHO) recommendations regarding regular physical activity (RPA), decreasing in frequency or number of days a week, (3.7 d/w to 2.9 d/w; *p* < 0.01) and duration (50.5 min/d to 45.2 min/d; *p* < 0.001) with age, especially for those with higher Gross Motor Function Classification System (GMFCS) mobility levels. Obesity slightly correlates with session duration (ρ = −0.207; *p* < 0.05), not mobility limitations. Gender has no significant impact on mobility, communication, or physical activity, while age affects variables such as body mass index (BMI) and engagement (*p* < 0.01). A substantial proportion follows regular physical activities based on health professionals’ advice, with interest decreasing with age. To improve adherence, focusing on sports-oriented goals, group sessions, and games is recommended. These findings emphasize the importance of personalized programs, particularly for older individuals and those with greater mobility limitations.

## 1. Introduction

According to the latest recommendations from the World Health Organization (WHO) regarding physical activity (PA) and sedentary behavior (SB), it is advised that children and adolescents engage on average in at least one hour per day of moderate-to-vigorous intensity aerobic PA. For adults, the recommendation is to participate in at least 150–300 min/week of moderate-intensity aerobic PA, or at least 75–150 min of vigorous-intensity aerobic PA, or a combination of both to achieve substantial health benefits [1]. The American College of Sports Medicine (ACSM) guidelines [2] provide more specific instructions, suggesting a frequency of 5 days/week of 30 min of moderate exercise or 3 days/week of 20 min of vigorous exercise to improve cardiorespiratory fitness. Furthermore, they recommend participating in muscle-strengthening activities involving major muscle groups for at least two days a week. Although these recommendations do not mention people with cerebral palsy (CP), nothing suggests a priori that these requirements should be different for them.

Regular physical activity (RPA) refers to planned, structured, repetitive PA, aimed at enhancing individuals’ health. When children and adolescents with typical development engage in RPA, it leads to improved physical fitness, cardiometabolic health, bone health, mental well-being, cognitive outcomes, and reduced adiposity. Similarly, in healthy adults, RPA significantly reduces the risk of developing cardiopathies, diabetes, high blood pressure, and breast and colon cancers [3]. Given these positive outcomes, increasing RPA among individuals with CP from childhood through adolescence may play a crucial role in enhancing both their morbidity and mortality rates. In children [4,5,6] and adolescents with CP [7,8] RPA has demonstrated benefits for motor functions, including enhancing gait speed [9,10], spasticity [11], balance [12] and muscle strength [10,13]. Therefore, encouraging and supporting RPA in this population can lead to substantial improvements in their overall health and well-being.

For people with CP, reaching the recommended guidelines for PA can pose many challenges; however, the encouraging aspect is that even engaging in PA below the recommended levels can still produce health benefits. The authors of Ref. [14] have formulated valuable guidelines pertaining to the frequency, intensity, duration, and types of activities that can serve as a helpful framework for prescribing RPA to individuals with CP. These recommendations have been developed based on a comprehensive review and analysis of relevant literature, expert opinions, and extensive clinical experience. Furthermore, they are aligned with the guidelines set forth by the ACSM [2].

-Frequency refers to the number of sessions conducted per week. Ideally, aiming for at least 3–5 sessions per week with a 24–36 h-recovery period in between is recommended to improve or maintain cardiorespiratory fitness. However, for people with CP who are very deconditioned, a more suitable approach would be to initiate with 1–2 sessions per week and gradually progress as they adapt to the physical demands [2,14,15].-Intensity is a crucial aspect linked to the effort exerted during PA, often expressed relative to maximal heart rate or heart rate reserve (HRR). HRR represents the difference between an individual’s measured or predicted maximum heart rate and their resting heart rate and/or peak oxygen consumption. For effective results, the intensity should generally align with the recommendation of exceeding 60% of the maximum heart rate or exceeding 40% of HRR throughout training, which is considered as a moderate or a vigorous exercise [16];-Time denotes the duration of each training session, which should be at least 20 min for aerobic workouts, and conducted consistently for a minimum of 8 to 16 consecutive weeks, depending on the chosen frequency;-A diverse range of exercises or types of activities can be adapted to suit the specific conditions and abilities of individuals with CP. These activities may include sports, running, muscle-strengthening exercises, and others, as long as they align with the recommendations provided by the therapist and are tailored to meet the individual’s unique needs and capabilities.

In the context of resistance training for adolescents with typical developments, Stricker and colleagues [17] have suggested that strength gains can be achieved through various types of resistance training, with a minimum duration of 8 weeks and a frequency of 2 to 3 times a week [18]. While some studies have indicated a duration period ranging from 10 to 16 consecutive weeks, it is important to note that longer training periods can lead to additional benefits. To further improve motor functions, adding weight to exercises has been found to be effective [19]. This practice has also shown positive results in improving bone health [20] and mineral density [21] in children with CP. Additionally, complementary techniques such as functional electro-stimulation (FES) during strengthening exercises [22] and cycling [23], as well as a technique involving whole-body vibration during stretching exercise [24], have shown promising outcomes.

The present study aims to address the following main questions:What is the percentage of individual with CP who adhere to recommendations for RPA in terms of frequency and duration of training sessions?Is there any loss of interest in physical activity over time among people with CP?If so, at what stage does this declining interest in physical activity typically occur?Are there specific mechanisms, tools, or strategies that could effectively engage people with CP in regular physical activity over an extended period?

The study also aims to explore secondary research goals, which include investigating the impact of SB in this population, understanding their exercise preferences, and identifying factors that could contribute to increased adherence to physical activity programs.

By addressing these research questions and secondary goals, the study seeks to contribute valuable insights into the physical activity patterns, motivations, and potential strategies to promote a more active lifestyle among individuals with cerebral palsy. The study endeavors to provide a comprehensive understanding of the factors influencing physical activity participation within this population, which can inform the development of targeted interventions and support programs.

## 2. Methodology

### 2.1. Study Design

This study adopts a cross-sectional study, descriptive, and analytical perspective. It focused on three regions of Spain: Andalusia (southern region), Madrid (central region), and Galicia (northwestern region). The research involved collaboration with Non-Governmental Organizations (NGOs) representing individuals with CP. Data collection using a specific questionnaire took place between May 2022 and October 2022.

### 2.2. Participants

People diagnosed with CP who regularly attend primary or secondary schools, special education schools, and specific centers that attend this population, were recruited for this study. A total of 177 individuals willingly provided informed consent, expressing their agreement to actively participate in this study. Out of those, 170 respondents, aged between 4 and 66, provided complete and accurate responses to the survey. The sample size was determined to achieve a precision error of ±7.5%, with a confidence interval of 95%, assuming maximum variability (*p* = 0.5) in a large population according to Cochrane’s formula [25].

### 2.3. Ethics Committee

This research has been approved by the Galician Research Ethics Committee, with the reference number: 2020/597, in 5 February 2021.

### 2.4. Procedure

To achieve the objectives of this study a specific survey was developed. The questions underwent verification and testing by professionals and by a reduced group of participants to ensure comprehensibility and clarity.

The survey consists of a 36-item questionnaire, including multiple-choice, open-ended, numerical, and Likert scale-type questions (refer to Appendix A). The participants, families, and caregivers were given the questionnaire to complete. When needed, participants received assistance from caregivers, family members, and/or the staff of the respective centers to ensure the questionnaire’s completion. The questions were grouped into several main aspects, summarized in Table 1, which include: (1) General demographic questions, such as age, height, body mass index, and sex; (2) Mobility level based on GMFCS [26]; (3) Information about PA (type, frequency, intensity), and activities of daily living (ADL)—which can be used to identify SB; (4) Personal beliefs regarding PA, motivation, preferences and the use of technological gadgets; and (5) Communication skills assessment through the Communication Function Classification System (CFCS) [27]. This system allows the classification of an individual’s daily communication into one of five levels based on the effectiveness of their communication.

Regarding the age of the participants, the sample was stratified into four groups, following the World Health Organization (WHO) recommendations (https://www.who.int/news-room/fact-sheets/detail/physical-activity, accessed on 20 March 2020): (1) Early childhood (younger than 5 years old); (2) Childhood and adolescence (ages 5 to 17); (3) Adults (ages 18 to 64); (4) Elderly (older than 65).

### 2.5. Analysis

The data obtained from the study were subjected to descriptive analysis. Qualitative variables were expressed with frequency and percentage, while quantitative data were described with mean (standard deviation), median, and range (minimum and maximum). To assess the normal distribution of the sample’s age, the Kolmogorov–Smirnov test was applied. The alternative hypothesis (*p* < 0.01), was assumed, indicating that the sample did not follow a normal distribution.

Considering the non-normal distribution of the sample, non-parametric tests were used to analyze the possible relationships between variables. The Spearman correlation was employed for quantitative variables, while the U Mann–Whitney or Kruskal–Wallis tests were used to determine differences in medians. For qualitative variables, the chi-square test was applied. The significance level for all hypothesis tests was set at *p* < 0.05.

The data analysis was conducted using SPSS v.27 software, which facilitated the application of the statistical tests and allowed for a comprehensive examination of the relationships between the study variables.

## 3. Results

### 3.1. General Population Description (Block 1)

Out of the initial 177 participants who completed the survey, 7 individuals did not provide complete responses and were subsequently excluded from the analysis. Therefore, the final sample for analysis consisted of 170 participants. Table 2 presents the sample description according to sex and age group. The sample comprises an equal distribution of 50% males and 50% females.

Due to the limited number of participants younger than 5 and older than 65 in the sample (only 4 individuals in each group), we decided to combine them into 2 larger groups for analysis. The final groups were categorized as follows: participants under 18 years old formed the “U18 group”, and those over 18 years old constituted the “O18 group”. This grouping allowed us to have more meaningful and robust comparisons while ensuring statistical reliability, given the small sample sizes in the extreme age ranges.

#### Body Mass Index (BMI)

The BMI [28] is a statistical index that combines an individual’s weight and height to estimate body fat. As per ACSM guidelines, individuals are classified into categories such as underweight, normal weight, overweight, or obese based on their BMI values, which also take gender and age into account. Figure 1 shows the collected BMI from the study participants and depicts the corresponding percentiles associated with age and gender. The number of participants falling on each side of the percentile curves, is detailed in Table 3, which also provides an overview of the percentages corresponding to the four BMI categories.

In general, it was observed that 77 participants (45.3%) were classified as overweight or obese, while 10% fall into the underweight category. Notably, there were no significant differences in BMI between genders, and the percentages of obesity were found to be similar in both male and female groups. However, we did identify some variations in the overweight category, particularly among participants over 18 years old (O18 group), where a higher percentage of individuals were categorized as overweight compared to the under 18 years old (U18 group). In the underweight category, the males in the U18 group exhibited the highest percentage.

### 3.2. Mobility and Communication Skills (Block 2)

Table 4 provides a summary of the results obtained from the questions within block 2. The data is presented separately based on the participants’ gender and age.

The majority of participants (*n* = 63, 37.1%) exhibited mobility level V, which entails wheelchair transportation in all environments. They experience limitations in maintaining antigravity head and trunk postures, as well as controlling leg and arm movements. The second most prevalent group was level I (*n* = 57, 33.5%), characterized by independent locomotion both indoors and outdoors. From mobility levels II to IV, 49 people reported using assistive devices during PA (primarily walkers, hoists, or DAFO-type orthoses).

The analysis of communication skills reveals a similar pattern, with the largest groups positioned at the classification’s extremes. A total of 58 individuals (34.1%) demonstrated independence and proficiency in alternating between sender and receiver roles in information exchange across most environments (level I). In contrast, 46 participants (27.1%) faced challenges in effective communication, even with familiar individuals (level V). Despite the difficulties observed in this latter group, only 15 participants acknowledged using communication assistive tools (primarily symbol boards). Notably, 30% of the sample believed that their communication abilities or their communication assistive devices could be influenced by their engagement in physical activities.

The mobility level exhibits a direct and robust correlation with participants’ communication proficiency (*p* < 0.001). Furthermore, this mobility level is associated with participants’ age, wherein older individuals tend to have lower gross motor function. It is evident that the utilization of communication assistive technology is influenced by participants’ reported communication level (*p* < 0.01).

### 3.3. Physical Activity (Block 3)

The survey has two parts. The first is common for all interviewees, and the second is only targeted at those who usually practice any kind of PA and answered yes to question Q11. We found that a high number of participants said yes to this answer (148, 87.1%), without significant differences in biological sex (males 89.4%, females 84.7%). This section contains the results associated with this subset of participants.

To begin with, our initial focus is on identifying the motivations behind engaging in physical activity, as addressed by questions Q25 and Q26 within the questionnaire (Table 5). In aggregate, 68.6% of responses indicate that participation in PA was recommended by professionals. When examining these responses across age groups, a notably higher percentage emerges among those under 18 (U18 group) at 89.5%, compared to the other age group (62.9%). This widespread recommendation for exercise might substantiate the prevalent dominance of rehabilitation/treatment as a choice, comprising 61% of responses (45.8% for the U18 group and 71.3% for the over 18 (O18) group). In contrast, a mere 14 participants (9.6%) selected ‘personal choice’ as their rationale.

In addition to rehabilitation, which a significant portion of interviewees undertake at NGO facilities (60.1%) or specialized centers (22.3%), they frequently engage in supplementary activities (questions Q18, Q23, and Q24). These activities include:-Walking or Trekking in Open Spaces: This is pursued by 59 individuals;-Aquatic Activities: These involve hydrotherapy or simple swimming and are practiced by 41 participants;-Adapted Sports: A group of 30 participants partake in sports like boccia, wheelchair basketball, soccer, athletics, martial arts, shooting, and sailing;-Gym Workouts: 12 individuals frequent the gym, where they often utilize adapted pedals and treadmills;-Equine Therapy: 8 participants find benefit in equine therapy;-Dancing: A smaller subset of 3 individuals engage in dancing.

These activities collectively represent the diverse range of physical pursuits among the interviewees.

A notable majority of individuals have a positive inclination towards engaging in physical activity (76.7%) (Table 6). Both male and female participants in the under 18 age group (U18) exhibited average scores of 3.8 and 4.0, respectively (on a scale of 1 to 5, where 1 represents “Do not like at all” and 5 represents “Like very much”). For the over 18 age group (O18), the scores were slightly higher at 3.9 for males and 4.1 for females. It is worth highlighting that no statistically significant differences were observed between age groups (O18, U18) and genders. Given that a substantial portion of the physical activity is tied to rehabilitation exercises, these findings underscore a general affinity towards PA, with no discernible waning sentiment over time.

Figure 2 depicts the accumulated weekly time in minutes across different age groups for three levels of physical activity (PA) intensity as recommended by either the WHO or the ACSM guidelines for moderate to vigorous exercise. The majority of participants primarily engage in light-intensity PA (58.8%). Notably, the under 18 age group (U18) exhibits a higher percentage of individuals involved in moderate-to-vigorous exercise (50%) in comparison to the over 18 age group (O18) at 36%. Interestingly, no participants were found engaging in vigorous PA beyond the age of 30. Consequently, only 17.6% of interviewees met the recommendations in terms of both accumulated time and intensity.

The ACSM recommends 3 to 5 exercise sessions per week, with each session lasting a minimum of 20 to 30 min, depending on the intensity. Our findings indicate an average of 3.1 sessions per week (SD = 1.72) for all participants, with an average session duration of 45.5 min (SD = 14.78). When comparing different age groups, the under 18 group (U18) had an average of 3.7 sessions per week (SD = 1.8), while the over 18 group (O18) had 2.9 sessions per week (SD = 1.8). This difference was statistically significant (*p* < 0.01). Notably, variations were also observed in session duration between these age groups. The U18 group had an average duration of 50.5 min per session (SD = 7.6), whereas the O18 group had an average duration of 45.2 min (SD = 19.3), with a statistically significant difference (*p* < 0.001).

When integrating these results with the data presented in Figure 2, it is evident that accrued exercise time diminishes as age increases, with an approximate decline of −2.5 min per year per week. The confidence interval for the regression slope is [−3.6, −1.4], indicating the direction and uncertainty of this relationship.

We found that the number of weekly exercise sessions is directly correlated with the duration of these sessions (ρ = 0.206; *p* < 0.01). In other words, there is a positive connection between the number of sessions they undertake each week and the duration of those sessions.

However, as elucidated in the introductory section, even a scaled-down PA program yields reported benefits to population health [14]. With a minimal PA regimen of 3 sessions per week, each lasting 20 min and involving moderate or vigorous exercise, a notable 39.7% of respondents adhered to this basic framework. The intensity of PA emerges as the pivotal factor curbing further escalation of this percentage.

### 3.4. Other Daily Activities (Block 4)

When examining the time allocated to various activities, the average weekly hours dedicated to locomotion (inclusive of wheelchair propulsion, where applicable) was 6.37 h/week (SD = 6.62). In spite of the fact that interviewees spend long time, on average, in this activity, it does not help meet the WHO and ACSM recommendations, since it is considered as a light PA. However, the positive aspect is that people who even admitted not engaging in PA, do some kind of activity. Only 10 people out of 170 interviewees showed a SB [30], with 80% of them belonging to O18 group, and a 60% with mobility levels I–II–III.

Passive leisure activities averaged at 10.54 h/week (SD = 10.48), while time dedicated to instrumental activities of daily life (IADL) amounted to 1.49 h per week (SD = 2.51). Especially, there were no significant differences based on participants’ gender or age. However, it is worth mentioning that those in the childhood-adolescence range exhibited the highest passivity in relation to activity types.

### 3.5. Personal Beliefs and Preferences (Block 5)

The majority of participants hold a strong belief that engaging in RPA is highly advantageous for their overall health and well-being, with only one individual expressing a contrary opinion (response to question Q10). This sentiment is particularly prevalent among those under the age of 30, but it tends to diminish as age increases. Especially younger participants exhibit a more pronounced endorsement of physical activity compared to their older counterparts, with statistical significance (*p* < 0.01). Gender, on the other hand, does not exert a discernible influence on these convictions. Interestingly, there appears to be a negative trend, contributing to an increase in the variability of opinions concerning physical exercise as individuals age (Figure 3).

Table 7 compiles the primary challenges reported regarding engaging in exercise or elevating the frequency of exercise (Question Q12). The predominant factor inhibiting individuals with CP from participating in physical activity is the absence of interest (n = 97). Among other noteworthy responses are the lack of family support (n = 69), physical barriers (n = 56), and unavailability of adapted equipment (n = 52). Notably, these hindrances are not attributed to insufficient information or guidance (n = 18). It is noteworthy that comparable responses were received from both genders, reflecting consistency across the responses.

Question Q13 elicited a diverse range of responses regarding the requisites for augmenting exercise or physical activity. A recurring sentiment was the desire to be part of a sports team or a group sharing similar interests and characteristics, which was perceived as a motivating and enjoyable avenue for engagement. In conjunction with group participation, the absence of adapted venues and materials for various activities such as sports, leisure, rehabilitation, aquatic pursuits, and games was indicated. Participants called for the presence of proficient personnel capable of providing support for these endeavors. An essential criterion for these resources is their proximity to the residences, minimizing the need for excessive effort to access them.

A significant obstacle to increasing exercise or PA is the substantial financial commitment, as therapies, facilities, materials, transportation, and more, often come with a substantial economic burden. Alongside economic constraints, respondents noted the scarcity of time and motivation as contributing factors to their limited engagement in physical exercise.

Overall, the responses underscore the paramount importance of an appropriate environment and support for engaging in physical activity safely and tailored to individual needs. Furthermore, personal motivation and the camaraderie of others emerged as crucial factors in sustaining consistency and interest in maintaining an exercise regimen.

In Q34, participants were presented with the question: “What kind of activities that take place in your association’s center do you think could increase your motivation to PA?”. Given the open-ended nature of this question, a diverse array of responses was obtained. Upon analyzing these responses, it becomes apparent that participants have articulated specific suggestions that could substantially elevate their motivation and consequently enhance their daily lives.

Among the provided answers, a prevailing request is for group-oriented games and other leisure activities that are attuned to their unique needs. This desire for engaging in recreational pursuits that are adapted to their circumstances resonates significantly. Additionally, there is a recurrent call for the inclusion of adapted sports such as boccia, basketball, football, and dancing. Within the realm of sports, some participants express an inclination towards participating in competitive formats.

There is a notable interest in gaining access to adapted swimming facilities that would offer activities such as hydrotherapy. Beyond structured sports, there is an expressed yearning for outdoor activities and interactions with animals, exemplified by the mention of equine therapy.

When queried about their present motivation (Q27) compared to their motivation when they were younger, 80.4% of adults (O18) responded that their motivation for engaging in physical exercise had not diminished over time. The remaining 19.6% provided explanations for this change, which included factors such as a decline in physical conditions (44.4% citing reduced strength, mobility, and overall abilities), reduced motivation (27.8%), concerns about falling (7.1%), and other underlying health conditions.

In the context of exercising, individuals exhibit a preference for group settings (36.9%), followed by 26.9% opting for solo workouts, and the remaining 36.2% expressing no particular preference. An interesting dimension emerges when considering gender differences. Females lean towards group exercise as their primary choice (39.4%), while males slightly exhibit a lower preference for group settings (34.6%). This gender divergence is a noteworthy aspect that merits attention.

Given that over one-third of the population favors engaging in PA within group environments, which inherently involves the commitment of others, there arises an elevated risk within this segment of the population for discontinuing regular exercise practices (Figure 4). This consideration highlights the potential challenges in maintaining consistent physical activity levels for individuals who lean towards group exercise.

Both the environment and the way for the practice of exercise (alone or in group) increase with the age of participants (*p* < 0.01).

Lastly, Q35 investigates the impact of communication skills and support systems on the engagement in RPA (Table 8). The responses are reasonably evenly distributed among the provided options in the overall findings. However, noteworthy distinctions emerge when examining gender-specific patterns. In the adult group of females, the “No” option was predominantly selected, indicating a perceived lack of effect. In contrast, in the child and teenager group, the opposite response was favored, implying that communication skills and support systems do influence their engagement in regular physical activity. These nuanced gender and age differences underscore the varying perceptions and influence that communication skills and support mechanisms have on the practice of physical activity across different demographic segments.

### 3.6. Assistive Technology, Gadgets, and Applications for Promoting Exercise and Controlling its Intensity (Block 6)

On one hand, our investigation revealed that individuals commonly utilize conventional gym equipment for their PA routines. These include diverse items such as balls of varying sizes, parallel bars, dumbbells, adapted bicycles, and pedals.

More than 80% of the participants reported not requiring any assistive devices for their PA activities (Question Q29). Among those who do rely on assistive devices, dynamic orthoses, walkers, manual or powered wheelchairs, standing frames, and robotic devices like Lokomat or Innowalk were prevalent choices. Table 9 contains the percentage of use of these assistive technologies, where the walking devices are, notably, the most used by users. It is noteworthy that conventional gym elements such as large balls, parallel bars, weights, adapted bicycles, and pedals were also utilized.

Turning to the utilization of software/applications for monitoring or promoting PA (Question Q31), a mere 5% of respondents confirmed its usage. A small subset of participants employed smartwatches for PA monitoring or tablets to maintain focus during exercise. For individuals utilizing Lokomat devices for rehabilitation, the equipment integrates software that provides visual feedback through video display. Especially, the utilization of monitoring devices or applications and participants’ comfort perceptions did not display any discernible association with demographic variables or the nature of their PA engagement.

Concluding this section, Figure 5 illustrates that half of the participants deemed wearing a monitoring device unnecessary. The remaining half was divided between those who consider it beneficial and those who refrained from responding to the question.

## 4. Discussion

This section encompasses the interrelationships among the studied variables along with an overarching discussion. To enhance clarity, we have introduced several subsections to categorize these interactions.

### 4.1. BMI Index Dependence on Mobility Level and PA


The results have revealed an overweight or obese percentage of 45.3%. To provide context, these figures will be situated within both a national and international framework. Table 10 reflects the percentages of BMI categories from Table 3 and integrates 2020 data from the INE (Instituto Nacional de Estadística, www.ine.es: accessed on 10 June 2022) for the Spanish population. Generally, it is evident that the collected data exhibited higher proportions of obese and underweight individuals compared to the general population. It is worth noting that a scoping review conducted by Ref. [31] examining fat status among children and young individuals with CP found no disparity when contrasted with those without CP. Additionally, it is observed that children with spastic CP exhibit heightened visceral fat but no discrepancies in total body fat when compared to their typically developing counterparts [32].

This raises questions about whether the interviewees’ overweight and obesity could be attributed to limited mobility and whether the group with the most restricted mobility (level V) exhibits the highest BMI values. To address these queries, Figure 6 presents a bar plot illustrating the percentages of each BMI category, organized by mobility and gender. Notably, within the female cohort, participants at mobility levels II (50%) and III (30%) exhibit the highest obesity percentages. Among males, the most elevated percentages occur at levels IV (37.5%) and II (25%). When combining overweight and obesity categories, the female group with mobility level II still registers the highest proportion, with similar distributions observed for the other mobility levels. Among males, individuals at mobility levels IV and II continue to have the highest percentages of overweight and obese individuals. In summary, our analysis does not reveal any discernible correlation between mobility level and obesity/overweight status. Thus, within our sample, no significant BMI discrepancies are observed in relation to gross motor function level.

Contrary to common assumptions, BMI does not exhibit a significant relationship with participants’ engagement in regular physical activity, its intensity, or the objectives for which it is pursued. In contrast, the impact of professional recommendations for physical activity seems to influence BMI, with higher values observed among participants who did not receive such guidance (*p* < 0.05). Additionally, a connection is established between BMI and session duration, revealing an inverse proportionality (ρ = −0.207; *p* < 0.05). However, BMI does not display any correlation with the weekly frequency of physical activity or the duration of other non-PA activities.

### 4.2. Relationship between PA and Mobility Level

The engagement in any form of PA experiences a decline over time, evidenced by diminishing frequencies and durations of training sessions. For individuals with mobility levels IV and V, this downward trajectory exhibits statistical significance, with rates of −3.7 min/week×year and −2 min/week×year, respectively (refer to Table 11). The mobility level exerts an influence on PA (*p* < 0.01), with those in mobility level V exhibiting the lowest participation, averaging 115 min/week. In contrast, levels I and III demonstrate higher engagement, with mean weekly durations of 183 min and 197.5 min respectively.

The weekly accumulated time dedicated to PA depends on both session frequency and duration. Our observations reveal that both the number of sessions and their respective duration are statistically impacted by the participants’ mobility level. However, the most important factor contributing to the accrued time is the duration of the sessions (*p* < 0.01). A noteworthy negative trend is evident in mobility levels IV and V, which is reflected in both session duration and frequency. It is worth noting that when considering gender-specific groups, there were no discernible statistical distinctions detected when comparing individuals with the same mobility levels.

The groups characterized by more significant mobility challenges (GMFCS IV–V) demonstrate the lowest adherence to the WHO guidelines for PA, with compliance rates of 12% and 14.3%, respectively. These figures are trailed by level I (14.6%), followed consecutively by levels II and III, with the latter displaying a higher compliance rate of 41.7%.

The intensity of physical activity PA engagement represents an additional intrinsic factor that is influenced by mobility level (*p* < 0.05). Especially, there exists a positive correlation between intensity and mobility, indicating that individuals with lower mobility levels (I, II) tend to engage in more intense PA. This intensity is further impacted by the environment in which the PA is conducted (*p* < 0.01), as individuals with more pronounced gross motor function limitations are more likely to practice PA within their treatment centers.

### 4.3. Relationship between PA with Other Variables

The utilization of assistive devices during PA, engaging in group exercises, or pursuing higher-intensity workouts exhibit statistical associations with the time dedicated to passive leisure (*p* < 0.01).

The frequency of participants engaging in PA is indeed a determining factor for their increased locomotion time (*p* < 0.01). This implies that individuals who engage in PA more frequently tend to have longer periods of movement, including wheelchair propulsion.

### 4.4. Relationship between Personal Beliefs and Other Variables

Individual beliefs regarding the significance of engaging in PA appear to have a positive influence on both the duration of sessions and their frequency per week (*p* < 0.01). However, no evident association is found with participants’ BMI. These personal convictions align with the recommendations provided by professionals (*p* < 0.01) and harmonize with the intended purpose of the PA. In cases where the exercise serves a rehabilitative function, there exists a link with the perception that it is highly recommended (*p* < 0.01).

As elucidated earlier, a preference for engaging in exercise within group settings is evident. Nevertheless, as mobility deteriorates, particularly beyond level III where walking necessitates the use of assistive devices, the proportion of participants opting for the ’alone’ option especially escalates (Table 12).

Regarding the potential influence of communication skills or support systems on PA, responses were fairly evenly divided between those who believe that communication does not impact their exercise routine and those who believe it does (Table 13). A notable proportion of affirmative responses were recorded among individuals without communication difficulties (level I). Conversely, among participants at other communicative skills, particularly those with higher levels, the ‘No’ response was more prevalent.

### 4.5. Engagement in PA

Children with cerebral palsy (CP) exhibit a 30% lower engagement in (PA) compared to their typically-developing peers and are twice as likely to engage in (SB) [33]. Furthermore, more than 75% of individuals with CP, both children and adults, are observed to spend the majority of their waking hours in sedentary pursuits [14]. Encouraging adherence to (RPA) regimens and cultivating a healthy lifestyle, beginning at an early age, becomes pivotal in preventing the gradual decline in motor abilities throughout the lifespan [34], and promoting increased aerobic fitness in adulthood [2].

The findings indicate that motivation to engage in PA is not correlated with demographic variables, skills, or exercise-related factors. However, the interviewees have expressed a decline in their level of motivation after the age of 30 in terms of recognizing the recommendation for physical exercise among individuals with cerebral palsy CP. Environmental obstacles, insufficient personnel, and a dearth of adapted resources emerge as hindrances to greater PA engagement. Respondents offer potential solutions to augment this activity, including the expansion of group-based activities, an enhanced presence of specialized personnel, and the incorporation of adapted games.

Exercise programs should be tailored to the individual’s mobility capabilities. There exists a diverse array of activities that have demonstrated their health and fitness benefits, including walking (both freely and on an elliptical or treadmill), cycling, employing a feet ergometer, utilizing elastic bands, participating in sports such as swimming or hockey, engaging in video-game activities, and interacting with animals [35]. However, many of these options are not viable for individuals with severe motor disabilities, such as those classified at level IV and V in the GMFCS, who primarily spend extended periods in a seated position [36]. The systematic review conducted by Ref. [9] comprehensively examined various exercise interventions involving children with CP across GMFCS levels I–III. The review’s findings indicated that while exercise interventions exhibited potential benefits in terms of gait speed and muscle strength, they did not yield a statistically significant impact on gross motor function within the CP population. In a more recent study by Ref. [37], the focus shifted towards investigating the impact of power training on locomotion capacities in children with CP at GMFCS levels III–IV. The study’s outcomes strongly imply that power training contributes to enhanced walking capacities in this specific group. Especially, there appears to be a gap in the literature, as no studies investigating PA programs among individuals classified under GMFCS level V were identified.

Especially, certain research findings underscore the advantages of incorporating brief, light-intensity exercises to interrupt or replace sedentary behavior [38,39,40], as well as the merits of frequent daily bouts of physical activity [2] for enhancing overall health.

Prioritizing the promotion of (PA) emerges as a crucial endeavor to prevent a deterioration in participants’ health. Regardless of its intensity, (RPA) exerts a favorable impact on elements such as balance, muscle strength, and gait speed [9,10,13], with benefits correlating to both time commitment and frequency [41]. Furthermore, an individual’s motivation to engage in PA programs plays a pivotal role in their full participation. Especially, our study underscores how participants’ motivation shapes the frequency, intensity, and type of PA activities they partake in.

Collaboration between health professionals, individuals with (CP), and their families emerges as pivotal to establish a comprehensive lifestyle and foster adherence to physical activity (PA) programs [42]. These programs should encompass a suitable exercise regimen, mitigate fatigue, and preempt adverse events [43,44,45,46]. Broadly, exercise is regarded as central to enhancing mobility and fitness, with the availability of facilities, programs, and a supportive attitude constituting pivotal factors for success [47]. Expanding the duration of PA engagement beyond center-based activities could yield positive outcomes. Engaging in home-based exercise is likely to require family support [48] and a safe, well-structured program that does not necessitate the continuous supervision of a healthcare professional. Especially, structured home-based exercise programs, whether provided on paper or online, have proven effective in enhancing gait [49], overall mobility, and the performance of daily tasks [50,51].

Interventions encompassing participation in competitive or collaborative groups, incorporating music during exercise, or featuring dance-based activities have also demonstrated their effectiveness. For instance, engaging in activities like boccia or hockey using powered wheelchairs [52] has yielded enhancements in daily activity performance among adults [21], motor functions in children [53], and the overall quality of life for children, young adults, and general health conditions [19]. Notably, the incorporation of stimulating music during resistance exercises led to notable increases in muscle strength and walking speed [54]. Similarly, interventions involving dance-based exercises have yielded improvements in hip and ankle range of motion (range of motion (ROM)), stride length, walking speed, cadence, and step measures [55]. Cluttercuck et al. [56] developed a practitioner-led, peer-group sports intervention called ‘Sports Stars’, and parents and physical therapists perceived it as effective and acceptable for children with CP with sports-focused goals.

Additional solutions involve the utilization of virtual assistants and robotic platforms. These applications incorporate various sensors to monitor exercise quality and duration, enabling activities to be conducted at home and customized to uphold specific physiological parameters within an appropriate range [57,58,59,60,61,62]. It is conceivable that the adoption of these technologies could potentially shift the perceptions of interviewees concerning wearable devices and lead to increased usage.

### 4.6. Limitations of the Study

The sample size, determined with 177 individuals, had a precision of ±7.5% under the conditions of maximum variability (*p* = 0.5) and a confidence interval of 95%. To achieve a reduced precision of ±5% we would have needed to double the sample size.

We did not conduct an assessment of cognitive functions. Such an assessment could have provided a more precise picture of the population. We plan to address this aspect in future research.

## 5. Conclusions

This extensive study, with a larger sample size, supports previous findings of a 17.6% adherence among individuals with cerebral palsy to recommended physical activity guidelines. Positive aspects include a high engagement rate (87.1%) in regular physical activity (RPA), with only 5.8% exhibiting sedentary behavior. The decline in both duration and frequency of physical activity over time is particularly prominent in individuals with mobility levels IV and V. There is a connection between the frequency of physical activity and the time spent on locomotion during daily living activities.

Approximately 45.3% of participants demonstrate overweight or obesity, slightly surpassing general population rates. This is not correlated with the mobility level GMFCS, suggesting that obesity may be linked to reduced physical activity duration. Age emerges as a determinant factor, affecting variables such as BMI and the practice of physical activity.

The number of hours dedicated to instrumental activities of daily living does not influence physical activity engagement, while the number of hours of passive leisure is associated with a higher number and duration of physical activity sessions, especially among the younger population.

A notable 68% of participants engage in RPA based on recommendations from health professionals, and 76.7% do so for rehabilitation or treatment purposes. Despite widespread acknowledgment of RPA benefits, adults with cerebral palsy exhibit a declining interest in exercising, particularly around their 30s. To counter this trend, the study recommends programs focus on sports-oriented goals, offer group sessions with specialized personnel, incorporate games, and facilitate strategies to overcome barriers.

## Figures and Tables

**Figure 1 healthcare-12-00535-f001:**
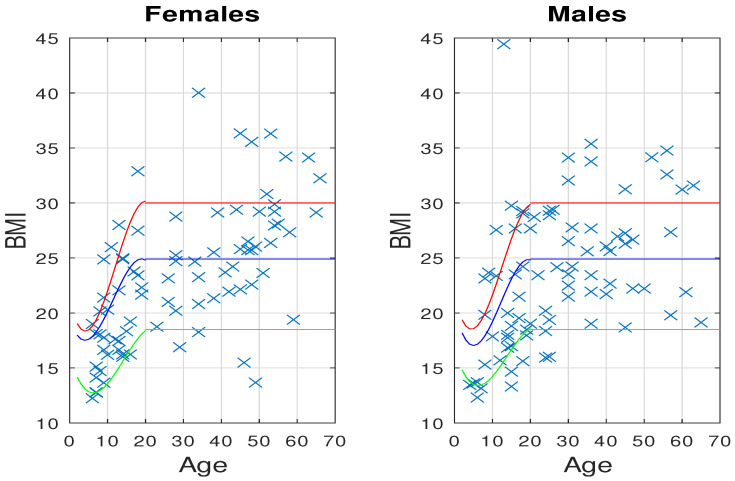
BMI index according to sex and age of the sample under study. The 5%, 85%, and 95% percentiles [29] are also drawn to show the limits for underweight (green line), overweight (blue line), and obesity (red line).

**Figure 2 healthcare-12-00535-f002:**
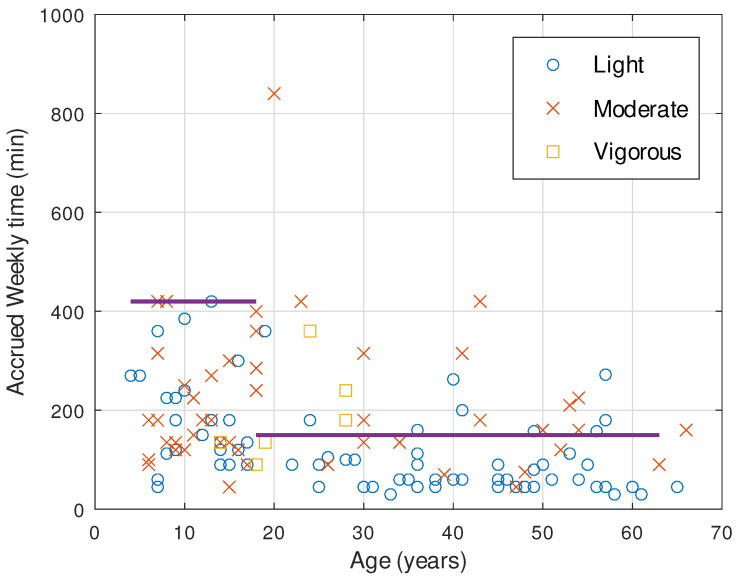
Accrued weekly time in the practice of PA for all participants and intensity type. The recommendation according to the age are also depicted in horizontal lines.

**Figure 3 healthcare-12-00535-f003:**
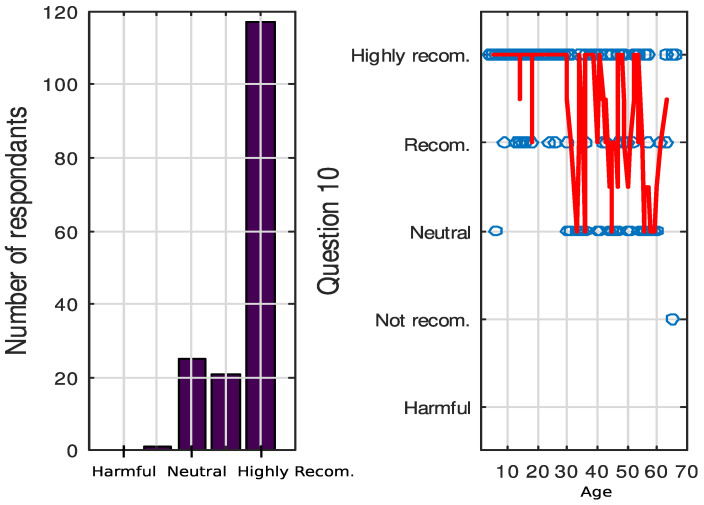
(**left**): answers to question 10 ‘Indicate whether you think it is advisable that people with CP practice PA regularly’. 5—Highly recommended, 1—Harmful. (**right**): the answer distribution over time for all participants. The red line is obtained by getting the median of 5-sample-length sliding windows.

**Figure 4 healthcare-12-00535-f004:**
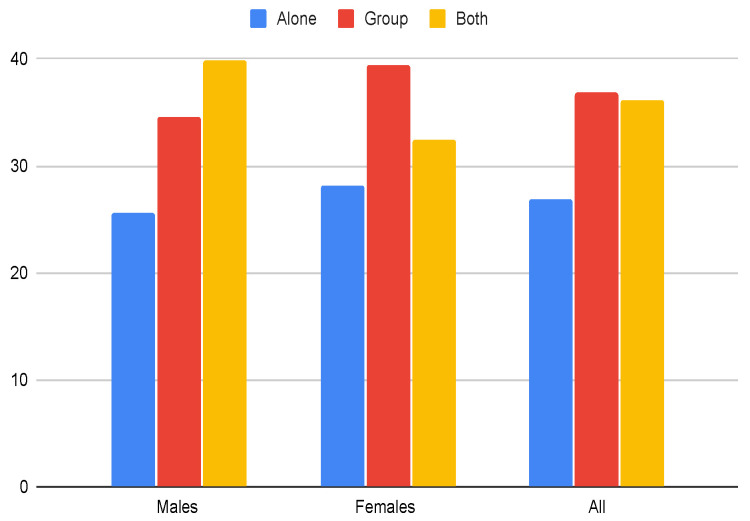
Question Q33 ‘How do you prefer to be physically active? Alone, Group, or Both’.

**Figure 5 healthcare-12-00535-f005:**
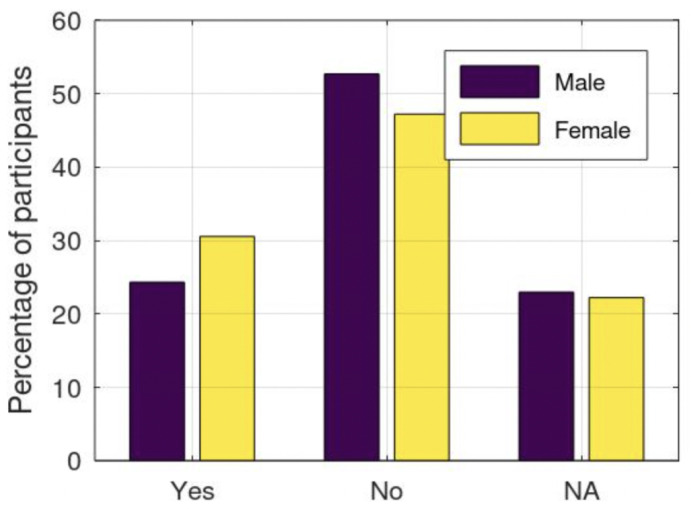
Responses to question Q19 ‘Do you use any assistive device for PA’ segregated by gender.

**Figure 6 healthcare-12-00535-f006:**
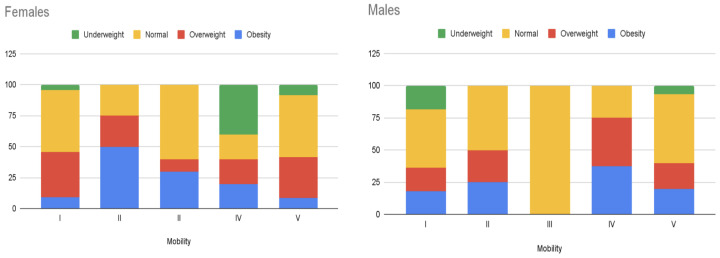
Bar plots showing the percentages of females and males within each BMI category: underweight, normal, overweight, and obesity.

**Table 1 healthcare-12-00535-t001:** Summary of the questionnaire’s items.

Block	Questions	Content
1	2–5	Demographic data collection about age, weight, height, and sex, including a label for identification. BMI was calculated with weight and height.
2	6–9	Mobility, communication skills, and the use of assistive technology.
3	11, 18–26	Aspects regarding physical activity (type, motivation, frequency, time, intensity, environment, purpose, recommendation)
4	14–16	Other daily activities (instrumental activities of daily life, leisure, and mobility in the community)
5	10, 12, 13, 27–28, 33–35	Personal beliefs and preferences about the practice of physical activities
6	29–32, 36	The use of technical gadgets or applications for promoting and controlling the intensity of the exercise.
7	1, 17	For control purposes. Item 1 is a label used for identification, and item 17 is a control question that allows participants to continue with the survey.

**Table 2 healthcare-12-00535-t002:** General population description.

Group	Males	Females	Total
Early childhood	1 (0.6%)	0	1 (0.6%)
Childhood and Adolescence	27 (15.9%)	34 (20%)	61 (38%)
Adults	56 (32.9%)	49 (28.8%)	105 (61.8%)
Older adults	1 (0.6%)	2 (1.2%)	3 (17.6%)
Total	85 (50%)	85 (50%)	170 (100%)

**Table 3 healthcare-12-00535-t003:** BMI for respondents segregated by age and biological sex.

BMI						
	**Overall n (%)**		**U18 Group n (%)**		**O18 Group n (%)**	
	**Males**	**Females**	**Males**	**Females**	**Males**	**Females**
Obesity	19 (22)	16 (18.3)	7 (25.9)	7 (19.3)	12 (20)	9 (17.6)
Overweight	19 (22)	23 (26.8)	1 (3.7)	3 (9.7)	18 (30.9)	19 (37.3)
Normal	37 (43.9)	39 (46.3)	15 (51)	20 (61.3)	22 (40)	19 (37.3)
Underweight	10 (12.1)	7 (8.6)	5 (18.5)	3 (9.7)	5 (9.1)	4 (7.8)

**Table 4 healthcare-12-00535-t004:** Classification of mobility and communication skills of participants by sex and range of age.

	Gender		Age Range		
**GMFCS Question Q6**	**Males**	**Females**	**Childhood & Adolescence** ** (U18)**	**Adults (O18)**	**Total**
I	27	30	33	24	57
II	8	12	8	12	20
III	3	11	6	8	14
IV	9	7	4	12	16
V	38	25	11	52	63
Total	85	85	62	108	170
**CFCS Question Q7**					
I	20	38	26	32	58
II	19	13	17	15	32
III	3	4	2	5	7
IV	15	12	9	18	27
V	28	18	8	38	46
Total	85	85	62	108	170

**Table 5 healthcare-12-00535-t005:** Q25–Q26 responses for identifying motivation.

		Gender				Age Range					
		**Male**		**Female**		**U18**		**O18**		**Total**	
		**Count**	**%**	**Count**	**%**	**Count**	**%**	**Count**	**%**	**Count**	**%**
Recommended	Yes	52	69.3	55	77.5	51	89.5	56	62.9	107	68.6
	No	23	30.7	16	22.5	6	10.5	33	37.1	39	31.4
Goal for PA	Rehab/Treatment	45	59.2	44	62.9	27	45.8	62	71.3	89	61
	Personal Choice	7	9.2	7	10	7	11.9	7	8	14	9.6
	Both	24	31.6	19	27.1	25	42.3	18	20.7	43	29.4

**Table 6 healthcare-12-00535-t006:** Responses to the Q19 question ‘If you exercise regularly, either as part of a rehabilitation program or as a personal choice, assess your motivation’.

	Males		Females		U18		O18		Total	
**Motivation**	**Count**	**%**	**Count**	**%**	**Count**	**%**	**Count**	**%**	**Count**	**%**
I don’t like it	1	1.4	2	2.9	0	0	3	4.8	3	2.2
I barely like it	10	14.3	10	14.5	9	17.7	11	17.5	20	14.4
I don’t mind it	7	10	2	2.9	5	9.8	4	6.3	9	6.5
I like it	30	42.9	26	37.7	19	37.2	40	63.5	56	40.2
I like it a lot	22	31.4	29	42	18	35.3	5	7.9	51	36.7

**Table 7 healthcare-12-00535-t007:** Question Q12 “What are the main difficulties in exercising or increasing the frequency of exercise?”.

	Males	Females	Total
Physical barriers in the environment	24	32	56
Lack of resources for sports practice near m residence	24	31	55
Not having the appropriate adapted sports equipment	26	26	52
Lack of information or advice	9	8	18
Lack of support from family members	31	38	69
Personal disinterest	43	54	97
Other reasons	16	19	35

**Table 8 healthcare-12-00535-t008:** Responses to the question Q35 ‘Do you think that your exercise or physical activity practice may be conditioned by your communication skills or support systems?’.

	Overall (%)		U18 Group (%)		O18 Group (%)	
	**Males**	**Females**	**Males**	**Females**	**Males**	**Females**
Yes	34.7	34.3	33.3	43.9	37.5	20.7
No	32.0	38.6	29.4	26.8	37.5	55.2
NA	33.3	27.1	37.3	24.3	25	24.1

**Table 9 healthcare-12-00535-t009:** Assistive technologies utilized during physical activity (PA), along with their frequency of use in the analyzed population.

Cathegory	Devices	Frequency (%)
Walking devices	Walkers, standing frames, parallel bars, treadmill	43.1
Positioning devices	Standing frames and cranes	8.6
Wheelchairs	Manual or powered	15.5
Orthoses	Splints, DAFO, wrist braces	15.5
Robotics	Exoskeletons, Lokomat	6.9
Others	Adapted bicycles, stabilo (skiing)	10.3

**Table 10 healthcare-12-00535-t010:** Comparison between BMI groups in the sample and the general population in Spain.

BMI								
	**Sample (%)**				**General Population in Spain (%)** ** (Source: INE, 2020)**			
	**U18 Group (%)**		**O18 Group (%)**		**U18 Group (%)**		**O18 Group (%)**	
	**Males**	**Females**	**Males**	**Females**	**Males**	**Females**	**Males**	**Females**
Obesity	25.9	19.3	25.9	19.3	10.4	10.2	16.5	15.5
Overweight	3.7	9.7	3.7	9.7	18.3	18.3	44.9	30.6
Normal	51	61.3	51	61.3	58.6	57.6	37.8	50.6
Underweight	18.5	9.7	18.5	9.7	12.6	14.0	0.8	3.3

**Table 11 healthcare-12-00535-t011:** Some PA features grouped by mobility level and gender. Significant trends (*p* < 0.01) are shown with a double asterisk.

	Males			Females			All		
**Mobility** ** Level**	**Freq** ** (days/w)**	**Duration** ** (min/session)**	**Trend** ** (min/w×y)**	**Freq** ** (days/w)**	**Duration** ** (min/session)**	**Trend** ** (min/w×y)**	**Freq** ** (days/w)**	**Duration** ** (min/session)**	**Trend** ** (min/w×y)**
I	3.6 (2.4)	53.4 (19.5)	−2.9	3.2 (1.3)	54 (22.7)	−0.5	3.3 (1.9)	53.7 (21.1)	−1.6
II	3.3 (1.4)	47.5 (6.1)	−1.1	4.2 (1.4)	41.9 (9.6)	−1.4	3.8 (1.4)	43.3 (8.5)	−1.1
III	3.3 (1.5)	60 (30.4)	1.2	3.8 (2.3)	49.4 (13.3)	−4.4	3.7 (2.1)	52.1 (17.)	−3.8
IV	3.6 (1.7)	40 (7.9)	−2.0	3.8 (2.1)	48 (12.6)	−4.9 **	3.7 (1.8)	42.9 (10.1)	−3.7 **
V	2.4 (1.5)	43.5 (10.4)	−2.6 **	2.7 (2.0)	41.5 (14.1)	−1.3	2.5 (1.7)	42.7 (12.0)	−2 **

**Table 12 healthcare-12-00535-t012:** Question Q33 ‘How do you prefer to be physically active? (Group activity would include playing sports such as boccia, wheelchair basketball, etc.)’. Units are in percentage.

	Males			Females			All		
**Mobility Level**	**Alone**	**Group**	**Both**	**Alone**	**Group**	**Both**	**Alone**	**Group**	**Both**
I	4.5	45.5	50	15.4	42.3	43.3	10.4	43.8	45.8
II	30	30	40	0	66.7	33.3	15.8	47.4	36.8
III	33.3	0	66.7	50	30	20	46.2	23.1	30.7
IV	33.3	44.4	22.3	40	20	40	35.7	35.7	28.6
V	35.3	29.4	35.3	42.9	33.3	23.8	38.2	30.1	30.1
Total	25.6	34.6	39.8	28.2	39.4	32.4	26.9	36.9	36.2

**Table 13 healthcare-12-00535-t013:** Question Q35 ‘Do you think that your exercise or physical activity practice may be conditioned by your communication skills or support systems?’. Units are in percentage.

	Males			Females			Overall		
**Communicative Skills Level**	**Yes**	**No**	**NA**	**Yes**	**No**	**NA**	**Yes**	**No**	**NA**
I	55	30.5	15	48.5	33.3	18.2	50.9	32.1	17
II	31.6	26.3	42.1	16.7	58.3	25.0	25.8	38.7	35.5
III	33.3	33.3	33.3	0	25	75	14.3	28.6	57.1
IV	28.6	35.7	35.7	22.2	44.4	33.4	26.1	39.1	34.8
V	17.4	30.4	52.2	38.5	30.7	30.8	25	30.6	44.4
Total	32.9	30.4	36.7	35.2	38	26.8	34	34	32

## Data Availability

The data collected in this research is hosted in the repository of the University of Seville https://idus.us.es/.

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
