# Peer review of "Physical Activity in Cerebral Palsy: A Current State Study"

_healthcare, 2024, doi:10.3390/healthcare12050535_

Round 1

Reviewer 1 Report

Comments and Suggestions for Authors

Thank you for study.   

In general, I think that every study on CP children is valuable, but in this study, the relationship between physical activity and the assistive devices used should also be determined. The classification of the asistive devices used (anterior walker, posterior walker, canedian, AFO, DAFO, etc.) can be added to the article. This is a very important factor and at least usage percentages can be added.

Were the validity and reliability tests of the 36-question survey conducted?

Was there a power analysis of the study?

Evaluation of cognitive functions is also very important. The importance of this can also be emphasized in the article for future studies.

Author Response

Dear reviewer, thank you for your valuable work.
Please find attached a file with the answers to your comments.

Reviewer 2 Report

Comments and Suggestions for Authors

Dear Authors,

First of all, I would like to congratulate the researchers for conducting this very interesting and innovative research. The object of study they have addressed is important for the target population (as well as for their socio-familial environment).

I have only a few suggestions for improvement:

ABSTRACT: Abbreviations are discouraged in this section, please remove them.
Incorporate some quantitative figures in the Results section.

INTRODUCTION: Correct and adequate.

METHODS: Authors should incorporate the calculation of the effect size of the final sample analysed.

RESULTS: Table 8 should be replaced by a figure.

DISCUSSION: The inclusion of Table 10, Table 11, Table 12, Table 13 and Figure 5 in the Discussion section is inappropriate. Please move section or delete.

Kind regards

Author Response

Dear reviewer, thank you for your work.
Please find attached a file with the answers to your valuable comments.

Reviewer 3 Report

Comments and Suggestions for Authors

Dear authors, nice hard work was done by you guys. However, few corrections I have mentioned in the attached PDF for improving it. All the best

Comments on the Quality of English Language

Good

Author Response

Dear reviewer, thank you for your work.
Please find attached a file with the answers to your comments.

Round 2

Reviewer 1 Report

Comments and Suggestions for Authors

Thank you for added information.